# MIAT Is an Upstream Regulator of NMYC and the Disruption of the MIAT/NMYC Axis Induces Cell Death in *NMYC* Amplified Neuroblastoma Cell Lines

**DOI:** 10.3390/ijms22073393

**Published:** 2021-03-25

**Authors:** Barbara Feriancikova, Tereza Feglarova, Lenka Krskova, Tomas Eckschlager, Ales Vicha, Jan Hrabeta

**Affiliations:** 1Department of Pediatric Hematology and Oncology, 2nd Faculty of Medicine, Charles University and Motol University Hospital, V Uvalu 84, 150 06 Prague, Czech Republic; barbara.feriancikova@gmail.com (B.F.); terezcern@gmail.com (T.F.); tomas.eckschlager@lfmotol.cuni.cz (T.E.); ales.vicha@lfmotol.cuni.cz (A.V.); 2Department of Pathology and Molecular Medicine, 2nd Faculty of Medicine, Charles University and Motol University Hospital, V Uvalu 84, 150 06 Prague, Czech Republic; lenka.krskova@lfmotol.cuni.cz

**Keywords:** neuroblastoma, *NMYC* amplification, lncRNA MIAT, cell metabolism

## Abstract

Neuroblastoma (NBL) is the most common extracranial childhood malignant tumor and represents a major cause of cancer-related deaths in infants. *NMYC* amplification or overexpression is associated with the malignant behavior of NBL tumors. In the present study, we revealed an association between long non-coding RNA (lncRNA) myocardial infarction associated transcript (MIAT) and *NMYC* amplification in NBL cell lines and MIAT expression in NBL tissue samples. MIAT silencing induces cell death only in cells with *NMYC* amplification, but in NBL cells without *NMYC* amplification it decreases only the proliferation. MIAT downregulation markedly reduces the NMYC expression in *NMYC*-amplified NBL cell lines and c-Myc expression in *NMYC* non-amplified NBL cell lines, but the ectopic overexpression or downregulation of NMYC did not affect the expression of MIAT. Moreover, MIAT downregulation results in decreased ornithine decarboxylase 1 (ODC1), a known transcriptional target of *MYC* oncogenes, and decreases the glycolytic metabolism and respiratory function. These results indicate that MIAT is an upstream regulator of NMYC and that MIAT/NMYC axis disruption induces cell death in *NMYC*-amplified NBL cell lines. These findings reveal a novel mechanism for the regulation of NMYC in NBL, suggesting that MIAT might be a potential therapeutic target, especially for those with *NMYC* amplification.

## 1. Introduction

In children with high-risk neuroblastoma (NBL), which represents the major cause of cancer-related deaths in infants, the overall five-year survival rate is approximately 40%. To date, there are no salvage treatment regimens known to be curative [1]. One of the most important molecular signs of high-risk NBL is an amplification of the *NMYC* oncogene. *NMYC* amplification or overexpression is associated with increased energy metabolism, rapid tumor growth, short survival rates, and unfavorable histology [2]. Moreover, it is also a well-established poor prognostic marker for NBL and strongly correlates with higher tumor aggressiveness and treatment resistance [2,3,4].

Long non-coding RNAs (lncRNA) are a heterogeneous group of RNA molecules more than 200 nucleotides in length that do not encode proteins of more than 100 amino acids [5,6]. LncRNAs contribute to various cellular processes in normal and disease states, and their expression is dysregulated in the majority of human cancers, including NBL [7]. LncRNAs can be regulated by MYC in different cancer types or control MYC expression, both at the transcriptional and post-transcriptional levels [8,9]. Myocardial infarction-associated transcript (MIAT) is a long intergenic non-coding RNA that carries out functions in the development of a variety of physiological and disease processes, including neuron development; schizophrenia; myocardial infarction; and various malignant tumors such as gastric cancer, pancreatic cancer, lung cancer, osteosarcoma, renal carcinoma, and laryngeal carcinoma [10,11,12,13,14,15]. MIAT in human cancers can act as a competitive endogenous RNA, play the role of miRNA sponge, regulate some signaling pathways, and affect some epigenetic modulators such as histone deacetylases and DNA methyltransferases [16,17,18,19,20]. Moreover, MIAT affects the alternative splicing of cell fate determinants and controls stem-cell commitment during neurogenesis and survival neurons [21]. The analysis of RNA sequencing data identified MIAT as one of the most abundant lncRNAs in NBL and as a candidate to module the activity of NMYC expression effectors [9,22].

We found that MIAT is upregulated in NBL tumor tissues and cell lines, and MIAT levels are associated with NBL and its *NMYC* status. Moreover, an in vitro assay showed that a knock-down of MIAT induced apoptosis and inhibited the proliferation, migration, and induced metabolic changes of NBL cell lines, particularly those with *NMYC* amplification. Our results and findings reveal a connection between lncRNA MIAT and NMYC, as well as c-Myc and its potential in NBL tumorigenesis.

## 2. Results

### 2.1. MIAT Expression Is Upregulated in Neuroblastoma Cell Lines and Tissues

We found a high MIAT expression in the NBL due to the microarray expression experiments in the NBL cell line (data not shown). According to the result, the expression of MIAT was measured in nine NBL cell lines: six lines with *NMYC* amplification (UKF-NB-4, UKF-NB-3, UKF-NB-2, UKF-NB-1, KELLY, IMR-32) and three without amplification (SK-N-AS, SH-SY5Y, SK-N-F1) [23,24]. The NMYC status in cell lines UKF-NB-1 and UKF-NB-2 was detected by fluorescence in situ hybridization (FISH) using a dual-color probe (Appendix A). The MIAT expression was also measured in HDFn as representative of non-tumor cells using qRT-PCR (Figure 1). Results show that the MIAT expression is noticeably higher in NBL cells with *NMYC* amplification than in cell lines without *NMYC* amplification (*p* < 0.01). Moreover, the MIAT expression in lines without amplification is lower than that in non-tumor HDFn cells (*p* < 0.01) (Figure 1A). All *NMYC*-amplified cell lines have an unequivocally higher NMYC mRNA expression than non-amplified cell lines, as measured by qRT-PCR assay (Appendix A). To determine whether MIAT is also expressed in tumor tissue and not only in cell lines, we analyzed the MIAT RNA expression using the RNA-ISH method in a total of 21 formalin-fixed paraffin-embedded (FFPE) tissues. We found a significantly higher MIAT positivity in high-risk NBL than in samples of lower risk or benign tumors, such as ganglioneuroma (*p* < 0.01) (Figure 1B–F). These results in cell lines and tumor tissues demonstrate that lncRNA MIAT is potentially associated with *NMYC* status (amplification and/or overexpression) in NBL and more expressed in NBL tumor tissues with a higher malignancy. Thus, MIAT can contribute to the malignant behavior of NBL.

### 2.2. MIAT Knockdown Inhibits Neuroblastoma Cell Proliferation and Induces Apoptosis in Cell Lines with NMYC Amplification

We found that MIAT is connected to *NMYC* status and is expressed in NBL tissue. Thus, we hypothesized that MIAT could exert a pro-tumorigenic function—namely, in NBL with *NMYC* amplification. To address this question, we first knocked down MIAT using siRNA in NBL cells. The transfection resulted in a significant suppression of the MIAT level compared to non-coding siRNA-transfected cells (Figure 2A). Cell viability assays at 24, 48, and 72 h after transfection indicated that knockdown MIAT inhibited cell proliferation and reduced the number of viable cells compared to non-coding siRNA transfected cells in all tested cell lines. An annexin V binding assay further demonstrated that MIAT silencing in NBL cells induced apoptosis only in cell lines with *NMYC* amplification. Interestingly, we found only growth inhibition in NBL cells without *NMYC* amplification (Figure 2B,C). These data show that MIAT is a significant positive regulator of NBL proliferation and that MIAT silencing markedly induces apoptosis in *NMYC*-amplified cells.

### 2.3. MIAT Regulates NMYC and c-Myc Expression at the mRNA Level

*MYC* oncogenes play a pivotal role in the process of NBL tumorigenicity. To explore the connection of NMYC and MIAT, we analyzed the NMYC and c-Myc expression after MIAT silencing in NBL cells with and without *NMYC* amplification. MIAT silencing significantly decreased the levels of NMYC mRNA (Figure 3C) and protein (Figure 3A,G) in *NMYC*-amplified UKF-NB-4 and KELLY cells; furthermore, it also decreased the c-Myc mRNA (Figure 3D) and protein (Figure 3B) expression in *NMYC*-nonamplified SH-SY5Y and SK-N-AS cells. Taken together, as shown in Figure 3A–D,G, MIAT silencing significantly reduced the NMYC and c-Myc expression at both the mRNA and protein levels. To confirm whether MYC expression is affected on the mRNA level, MIAT knockdown cells were treated by proteasome inhibitor MG-132 (1 μM for 4 h). Figure 3E and Appendix A show that MIAT knockdown after proteasome inhibition reduced the NMYC protein expression compared to the increased NMYC expression in the control group; thus, MIAT regulates NMYC at the mRNA level. Moreover, we tried to test the transcriptional activity of MYC after MIAT silencing. We used one of the known downstream targets of *MYC* oncogenes—ornithine decarboxylase (ODC1)—in both *NMYC*-amplificated and non-amplificated cell lines (UKF-NB-4, KELLY, SH-SY5Y, SK-N-AS). The results indicated that MIAT knockdown significantly decreased the ODC1 expression (Figure 3F). We then transfected NMYC overexpression plasmid into NBL cell lines without *NMYC* amplification; the transfection markedly increased the NMYC mRNA (Appendix A) and protein expression (Appendix A) and also MYC target gene ODC1 (Appendix A). However, the ectopic overexpression of NMYC did not affect the MIAT levels in transfected cell lines. Moreover, NMYC downregulation by siRNA silencing has no significant effect on MIAT expression (Appendix A). These results indicate that MIAT is an upstream regulator of NMYC and verify the connection between MIAT and *NMYC* amplification.

### 2.4. Downregulation of MIAT Causes Changes in Cell Cycle Distribution and Inhibits Neuroblastoma Cells Migration

We found that MIAT downregulation inhibited cell proliferation (Figure 2C and Figure 5B). Thus, we explored the role of MIAT in the cell cycle and migration. Consistent with the proliferation assay data, the cell cycle distribution indicates that MIAT knockdown results in a significant increase in cells in the G_0_/G_1_ phase in UKF-NB-4 cells, a significant decrease in the S phase, and a significant increase in cells in the G_2_/M phase in both the UKF-NB-4 and SH-SY5Y cells as compared to the controls (Figure 4). Similarly, NMYC downregulation significantly increases cells in the G_2_/M phase and significantly decreases cells in the S phase in UKF-NB-4 and KELLY cell lines (Appendix A).

Migration is an important ability of tumor cells and plays a key role in metastasis and cell survival. We performed a wound-healing assay as well as a xCELLigence system measurement. MIAT downregulation significantly reduced UKF-NB-4 and SH-SY5Y cell migration (Figure 5 and Appendix A). These results indicate that MIAT might act as a regulator of the cell cycle and promote the migration of NBL cells.

### 2.5. MIAT Knockdown Decreases Glycolytic and Respiratory Capacity of Neuroblastoma Cells

Since *MYC* oncogenes have been reported to induce the upregulation of cellular metabolic functions [25], we examined the cell energy glycolytic and respiratory activity. MIAT downregulation reduced the glycolysis function–extracellular acidification rate (ECAR) compared to the control group (Figure 6A). MIAT knockdown decreased the glycolysis, glycolytic capacity, and glycolytic reserve in both UKF-NB-4 and SH-SY5Y cells, compared to the group with non-coding siRNA (Figure 6B). The decrease in the glycolytic reserve after MIAT knockdown was significantly higher in UKF-NB-4 (*NMYC*-amplified cells) than in SH-SY5Y (*p* < 0.05) (Figure 6B).

The mitochondrial oxidative phosphorylation–oxygen consumption rate (OCR), significantly decreased in the MIAT siRNA group compared to the control group (Figure 7A). MIAT knockdown decreased basal and maximal respiration, ATP-associated OCR, proton leak levels, and spare respiratory capacity in UKF-NB-4 and SH-SY5Y cells compared to the cells transfected with non-coding siRNA (Figure 7B). These results demonstrate that MIAT downregulation reduces glycolytic and respiratory function in neuroblastoma cells.

## 3. Discussion

NBL is the most common extracranial childhood malignant tumor and represents the major cause of cancer-related deaths in infants. Thus, the identification of new strategies to improve current treatments is needed. LncRNA MIAT is a disease-associated lncRNA and is dysregulated in multiple disorders. MIAT is also a negative prognostic factor in different tumors [11,12,13,14,15]. However, the association and function of MIAT in NBL remains unclear. Our study demonstrates that MIAT expression is markedly elevated in human NBL cell lines—namely, with *NMYC* amplification—and in the FFPE tissue of high-risk NBL patients; furthermore, MIAT knockdown decreases the NMYC or c-Myc expression and induces apoptosis and metabolic changes. Consistent with our results, the RNA sequencing data from Rombaut et al. [9] showed that MIAT is one of the most expressed lncRNAs in NBL. Moreover, they described an association between MIAT as an upstream regulator of NBL drivers genes *NMYC* and *Phox2B*. More recently, and consistent with our findings, MIAT silencing also downregulated the transcription activity of MYC in endothelial cells [26]. We confirmed the association between MIAT, NBL, and NMYC expression in the cell lines and NBL FFPE tissue.

In the current study, we found that MIAT silencing induces apoptosis significantly only in cells with *NMYC* amplification, but only inhibits cell growth in *NMYC* non-amplified cells. Bountali et al. [22], consistent with our results, detected low-rate apoptosis in one non-amplified neuroblastoma cell line. This enhances the potential association between MIAT expression and *NMYC* amplification. In the FFPE analysis, we did not confirm the relationship between *NMYC* amplification and MIAT expression. That may be due to a too small sample size and the individual heterogeneity of the group. We detected elevated MIAT expression in the high-risk NBL group. MIAT silencing does not induce the inhibition of cell growth in the human non-tumor cell line HDFn (Appendix A). However, the MIAT expression level in HDFn is even higher than in NBL cell lines without *NMYC* amplification.

We also found that MIAT downregulation reduces the NMYC expression in *NMYC*-amplified cell lines and c-Myc expression in *NMYC* non-amplified cell lines. Moreover, MIAT downregulation decreases ODC1, a transcriptional target of *MYC* oncogenes [27] and the rate-limiting enzyme in polyamine biosynthesis. Indeed, elevated ODC1 (independent of *NMYC* amplification) is associated with reduced survival in NBL [28]. We then genetically manipulated two NBL cell lines by NMYC overexpression or knockdown and did not observe any significant changes in MIAT expression. These results, together with the findings of Rombaut et al. [9], suggest that MIAT acts as an upstream regulator of *MYC* oncogenes, but its expression is not directly regulated by *MYC* oncogenes. Our results for proteasome inhibiton indicate that MIAT affects NMYC expression, especially at the mRNA level. A more detailed study is needed to confirm the exact molecular mechanism of how MIAT is involved in the regulation of *NMYC/c-Myc* oncogenes in NBL. The reduction in the NMYC expression level promotes cell cycle changes, differentiation, and apoptosis and decreases energy metabolism [29]. Moreover, MIAT regulates numerous genes that are involved in the cell-cycle progression and genes that have a role in the early-stage neuronal differentiation and signaling [30]. Our data indicate that NMYC knockdown results in a significant decrease in the proliferation fraction in NBL cells. In addition, MIAT knockdown results in a significant arrest of cells in the G_0_/G_1_ phase only in amplified NBL cell lines (with high MIAT expression) compared to the control group or cells without MIAT overexpression. MIAT knockdown retards the G_0_/G_1_ to S transition in *NMYC*-amplified NBL cell lines. Thus, these results indicate a paired MIAT and *NMYC* oncogene cell cycle control.

Studies showed that *MYC* oncogenes are associated with the regulation of energy metabolism through the direct or indirect activation of genes involved in glycolysis, glutamine and fatty acid metabolism, and mitochondrial function [31,32,33]. MIAT knockdown decreases the glycolysis, glycolytic capacity, and glycolytic reserve in both amplified and non-amplified cells. Thus, our results show that MIAT is involved together with NMYC and c-Myc in cell cycle progression, cell proliferation, and energy metabolism in NBL cells.

Taken together, there is a clear connection between the MIAT expression profile and *NMYC* amplification/non-amplification in NBL cell lines. We suggest that MIAT is involved in NBL progression and could be a candidate as a novel tumor marker for diagnosis and NBL treatment, especially for those with *NMYC* amplification. Given that NMYC expression is closely related to the prognosis of NBL patients, we believe that reducing the MIAT expression is a potential strategy to improve the prognosis of high-risk NBL patients.

## 4. Materials and Methods

### 4.1. Cell Culture and Transfection

The UKF-NB-4, UKF-NB-3, UKF-NB-2, and UKF-NB-1 NBL cell lines were a present from Prof. J. Cinatl, Jr. (J. W. Goethe University, Frankfurt, Germany). The HDFn (human dermal fibroblasts, neonatal) was purchased from Thermo Fisher Scientific. NBL cell lines KELLY, IMR-32, SH-SY5Y, SK-N-AS, and SK-N-F1 were from ATCC. Cell lines were cultivated at 37 °C and 5% CO_2_ in Iscove’s Modified Dulbecco’s Medium (IMDM) with 10% fetal bovine serum (both Thermo Fisher Scientific, Waltham, MA, USA). For silencing, a smart pool of four siRNAs, designed to target all variants of MIAT (NCBI accession numbers NR_003491, NR_033319, NR_033320, NR_033321) and NMYC (NCBI accession numbers NM_001293228, NM_005378, NM_001293233, NM_001293231), were purchased from Dharmacon. The 25nM MIAT siRNA and 50nM NMYC siRNA were transfected into NBL cells using Dharmafect transfection reagent (Dharmacon, Lafayette, CO, USA).

### 4.2. Cell Viability Assay

To determine viable cells, the cells were incubated with 25 μL PrestoBlue^®^ Cell Viability Reagent (Thermo Fisher Scientific) for 30 min at 37 °C. The fluorescence was measured using an excitation wavelength of 560 nm and emission of 590 nm by SpectraMax^®^ i3x Multi-Mode Microplate Reader (Molecular Devices, San Jose, CA, USA). The xCELLigence RTCA DP Instrument (Hoffmann-La Roche, Basel, Switzerland) was placed in a humidified incubator at 37 °C and 5% CO_2_ was used for the real-time monitoring of cell proliferation. Twenty-four hours after transfection, the UKF-NB-4 and SH-SY5Y cells were seeded into wells of 16-well E-plates for impedance-based detection. The cell index was monitored every 30 min for 96 h and data were recorded by the supplied RTCA software.

### 4.3. Annexin V/DAPI Staining Assay

For the detection of apoptosis, Annexin V-Dy647 (Apronex) and DAPI (4’,6-Diamidino-2-Phenylindole) (Thermo Fisher Scientific) staining were used as described in the manufacturer’s instructions. Briefly, Annexin V/DAPI Staining Assay was utilized 24, 48, and 72 h after siRNA transfection. The cells were resuspended in 100 μL of a binding buffer with 1 μL of Annexin V-Dy647 and 1 μg/mL of DAPI. After 15 min of incubation at room temperature, the cells were measured by flow cytometry using BD FACSCelesta (BD Bioscience, Franklin Lakes, NJ, USA), and data analysis was performed by Flowlogic software (Inivai Technologies, Mentone Victoria, Australia).

### 4.4. Cell Cycle Analysis

The cell cycle was analyzed using the Click-iT™ EdU Alexa Fluor™ kit (Thermo Fisher Scientific) according to the manufacturer’s protocol 48 h after transfection. The cells were cultured for 2 h with a 10 μM of EdU (5-ethynyl-2′-deoxyuridine). The cell cycle was measured by flow cytometry–BD FACSCelesta (BD Bioscience), and data were analyzed by Flowlogic software (Inivai Technologies).

### 4.5. RNA Extraction and Reverse Transcription-Quantitative Polymerase Chain Reaction (RT-qPCR)

The total RNA from cultured cells was extracted using PureLink^®^ RNA Mini Kit (Ambion) according to the manufacturer’s protocol. The extracted RNA’s quality was measured by the NanoDrop One spectrophotometer (260/280 nm ratio) (Thermo Fisher Scientific). Complementary DNA was synthesized from 1 ng of RNA using Generi Biotech Reverse Transcription Kit according to the manufacturer’s instructions (Generi Biotech, Hradec Kralove, Czech Republic). The primers and probes were designed and produced by Generi Biotech. The primer sequences are shown in Appendix A. The expression levels of target genes and control gene (POLR2A) were analyzed with real-time PCR on a QuantStudio 3 Real-Time PCR System (Thermo Fisher Scientific). The relative differences in gene expression were expressed as fold change and were obtained with the 2^−ΔΔCt^ method [34] (REST software).

### 4.6. Cell Migration

The migration of UKF-NB-4 and SH-SY5Y cells was determined using an xCELLigence RTCA DP Instrument (Hoffmann-La Roche) and wound-healing assay. For xCELLigence analysis, the cells were seeded 24 h after transfection in 100 μL of serum-free medium in the upper chamber with 8 mM pore size, while the lower chamber contained medium with 10% serum. Cell migration was monitored every 30 min for 24 h and data were recorded by the supplied RTCA software. For the wound-healing assay, NBL cells were seeded in a 35mm dish and allowed them to reach 70% confluence. The line was drawn across the dish’s surface using a 1 mL sterile plastic tip one day after transfection. Images were captured at the same field 24 and 48 h after the wounding by microscope Olympus IX51 (Olympus, Shinjuku, Japan). The migration distance was measured by ImageJ software program (National Institutes of Health, Bethesda, MA, USA).

### 4.7. Cellular Energetics

The cell respiratory and metabolism energy functions of silenced and control cells were determined using Seahorse XFp Extracellular Flux Analyzer (Agilent Technologies, Santa Clara, CA, USA). We measured the oxygen consumption rate (OCR), as an indicator of mitochondrial respiration, and the extracellular acidification rate (ECAR), as an anaerobic glycolysis indicator. The cells were reseeded in a special Seahorse XFp Cell Culture Miniplate one day before assay and one day after transfection. The XF assay medium was supplied according to the manufacturer’s protocol. Cell Mito Stress Test was performed using the following final concentration of FCCP injection after titration 0.125 μM FCCP (UKF-NB-4), 2 μM FCCP (SH-SY5Y). A Glycolysis Stress Test was performed using the following final concentration of 2 μM oligomycin injection after the titration of both UKF-NB-4 and SH-SY5Y cell lines (all, Agilent Technologies).

### 4.8. SDS-PAGE and Western Blot Analysis

All the cells were collected and lysed in RIPA buffer (Sigma-Aldrich, St. Louis, MI, USA) containing protease inhibitors (Roche Diagnostics, Basel, Switzerland). The protein concentration was measured using DC protein assay (Bio-Rad Laboratories, Hercules, CA, USA) according to the manufacturer’s instructions. Ten micrograms extracted proteins in NMYC amplified cells, 50 μg NMYC non-amplified cells, 30 μg for c-Myc detection were electrophoretically separated by SDS-PAGE electrophoresis using 4–20% TGX precast gel (Bio-Rad Laboratories). The separated proteins were transferred to a nitrocellulose membrane and blocked with 5% non-fat milk (Bio-Rad Laboratories) for 1 h at room temperature. The membranes were exposed to anti-NMYC rabbit monoclonal antibody and anti-c-Myc rabbit monoclonal antibody (both 1:1000; Cell Signaling Technology). Antibody against actin (1:1000; Sigma-Aldrich) was used as a loading control. Membranes were washed and exposed to Europium conjugated anti-IgG secondary antibodies (1:5000; Molecular Devices), and the antigen-antibody complex was visualized by SpectraMax i3x (Molecular Devices).

### 4.9. Tissue Collection and RNA In Situ Hybridization

Twenty-one FFPE specimens of neuroblastic tumors (15 NBL, 4 ganglioneuroblastoma, and 2 ganglioneuroma) were collected retrospectively from the archives of Department of Pathology and Molecular Medicine at the University Hospital Motol in Prague, Czech Republic. Sections of FFPE neuroblastoma tissues (4 μm) were processed according to the manufacturer’s description for RNAscope 2.5 assay (Advanced Cell Diagnostics, Newark, CA, USA) for lncRNA in situ detection of MIAT. Briefly, deparaffinized sections were hybridized to probes MIAT, followed by amplification by serial application of amplifiers, followed by peroxidase labels, and detection with Fast Red. The slides were counterstained with hematoxylin for 15 sec at room temperature. Expression of DapB and the housekeeping gene POLR2A were used as negative and positive control probes, respectively. Positive staining was determined by red punctate dots in the cells. The Histoscore (H-score) was calculated by a semi-quantitative assessment of the intensity of staining and the percentage of MIAT positive cells in FFPE NBL tissues, staining by RNA-ISH method. The range of possible scores was from 0 to 300.

### 4.10. Immunofluorescence Staining

The cells were plated onto cover slides in 24-well plates. The cells were fixed, permeabilized, and blocked according to the manufacturer’s protocol 48 h after transfection. The dilution of primary antibodies was 1:500 for cells with *NMYC* amplification and 1:50 for cells without amplification (Cell Signaling Technology, Danvers, MA, USA). The cells were incubated for 1 h in the dark with Alexa Fluor 647 Goat anti-rat IgG (Life Technologies, Prague, Czech Republic) at a dilution of 1:300. Afterward, the cells were stained with 2mg/mL Hoechst 33,342 and then with Actin-RED (Thermo Fischer Scientific). The slides were mounted with ProLong™ Diamond Antifade Mountant (Thermo Fischer Scientific). Images were acquired using the confocal microscope Leica SP8 (Leica Microsystems, Wetzlar, Germany).

### 4.11. Statistical Analysis

Data are shown as the mean ± standard error. All the experiments were independently repeated at least three times. ANOVA with the post hoc Tukey HSD test was utilized when comparing the treatments. The results from RT-qPCR were statistically compared using the REST 2009 software [35]. *p* < 0.05 was considered significant.

## Figures and Tables

**Figure 1 ijms-22-03393-f001:**
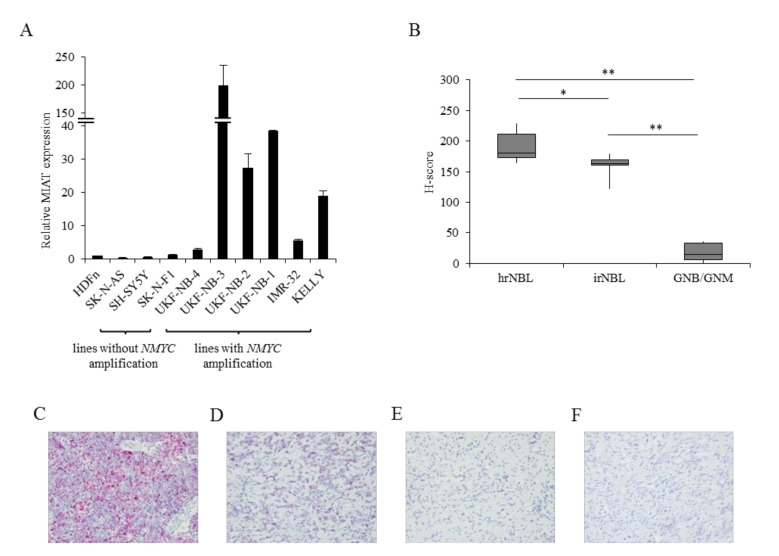
Myocardial infarction associated transcript (MIAT) expression in neuroblastoma (NBL) cell lines and tissues. (**A**) MIAT expression is significantly higher in NBL cell lines with *NMYC* amplification than in cell lines without *NMYC* amplification (*p* < 0.01) measured by RT-qPCR. (**B**) The box plot shows the distribution of the H-score in high- and intermediate-risk neuroblastoma and ganglioneuroblastoma/ganglioneuroma. The horizontal bar represents the median value of each score. MIAT expression in (**C**) high-risk formalin-fixed paraffin-embedded (FFPE) NBL tissue, (**D**) intermediate-risk neuroblastoma, (**E**) ganglioneuroblastoma, and (**F**) ganglioneuroma tissue measured by RNA-ISH method, (magnification, x400). Values are mean ± SD from three independent experiments. ** *p* < 0.01, * *p* < 0.05 (ANOVA with post hoc Tukey HSD Test).

**Figure 2 ijms-22-03393-f002:**
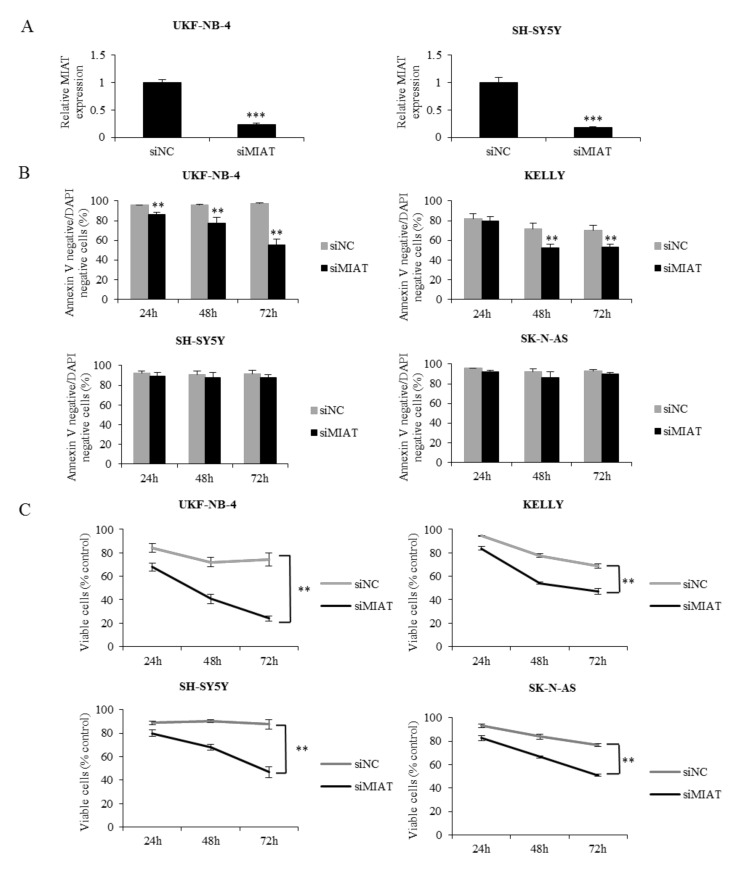
MIAT knockdown in human NBL cell lines induces apoptosis in lines with *NMYC* amplification compared to cells without *NMYC* amplification, where it only inhibits the growth. (**A**) The RT-qPCR bar graphs demonstrate the decreased MIAT expression levels in UKF-NB-4 and SH-SY5Y cells after siMIAT transfection. (**B**) The plot shows the effects of MIAT knockdown in NBL cell lines with and without *NMYC* amplification on the number of Annexin V and DAPI-negative, non-apoptotic cells. (**C**) Viable UKF-NB-4, KELLY, SK-N-AS, and SH-SY5Y cells after MIAT knockdown were measured by PrestoBlue at different time points. Values are mean ± SD from three independent experiments. *** *p* < 0.001, ** *p* < 0.01 compared with siMIAT/siNC group (ANOVA with post hoc Tukey HSD Test).

**Figure 3 ijms-22-03393-f003:**
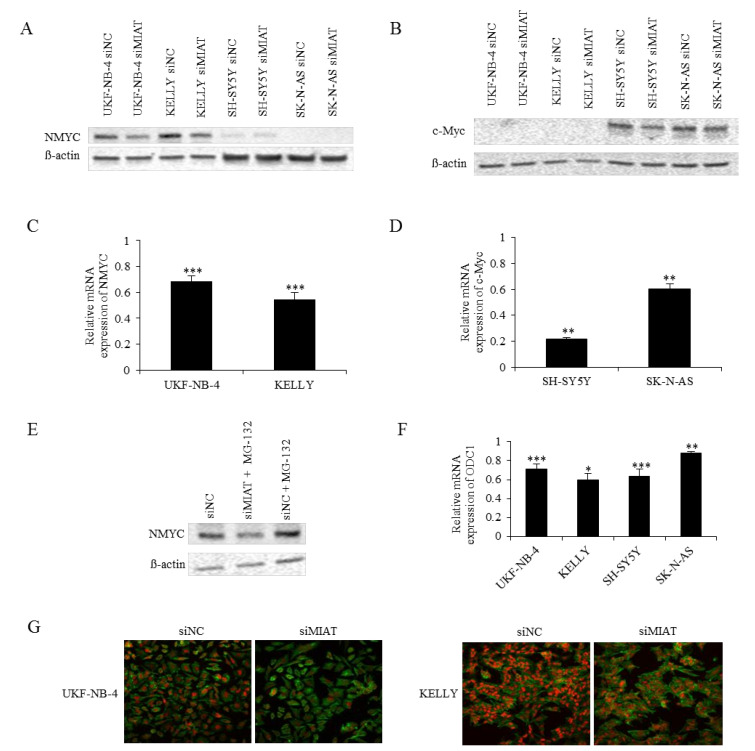
MIAT downregulation reduces NMYC expression in *NMYC*-amplified cell lines and c-Myc expression in *NMYC* non-amplified cell lines. (**A**) The NMYC and (**B**) c-Myc protein expression; (**C**) the NMYC, (**D**) c-Myc, and (**F**) ODC1 mRNA expression in siNC or siMIAT-transfected neuroblastoma cells. (**E**) The NMYC expression in siNC or siMIAT-transfected neuroblastoma UKF-NB-4 cells and simultaneous proteasome inhibition. (**G**) The NMYC expression after MIAT downregulation was also determined by immunofluorescence staining- NMYC- red, actin- green (630× magnification). The image shows representative data of three independent experiments. Values are mean ± SD from three independent experiments. *** *p* < 0.001, ** *p* < 0.01, * *p* < 0.05 as compared to the siMIAT/siNC group (REST 2009).

**Figure 4 ijms-22-03393-f004:**
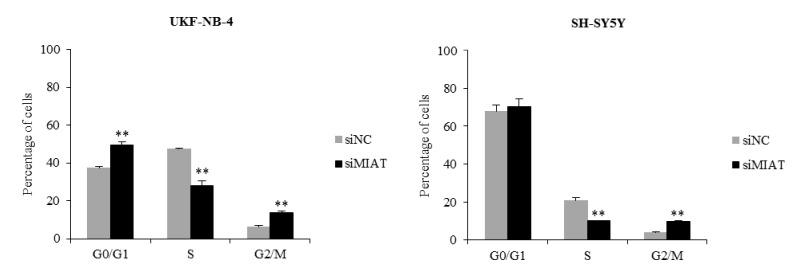
MIAT knockdown by siRNA increases cells in the G_0_/G_1_ phase in *NMYC*-amplified cells, decreases the number of cells in the S phase, and arrests cells in the G_2_/M. The percentage of cells in the G_0_/G_1_, S, and G_2_/M phases in UKF-NB-4 and SH-SY5Y cells transfected with siMIAT and siNC was determined by flow cytometry. Values are mean ± SD from three independent experiments. ** *p* < 0.01 as compared to siMIAT/siNC group (ANOVA with post hoc Tukey HSD Test).

**Figure 5 ijms-22-03393-f005:**
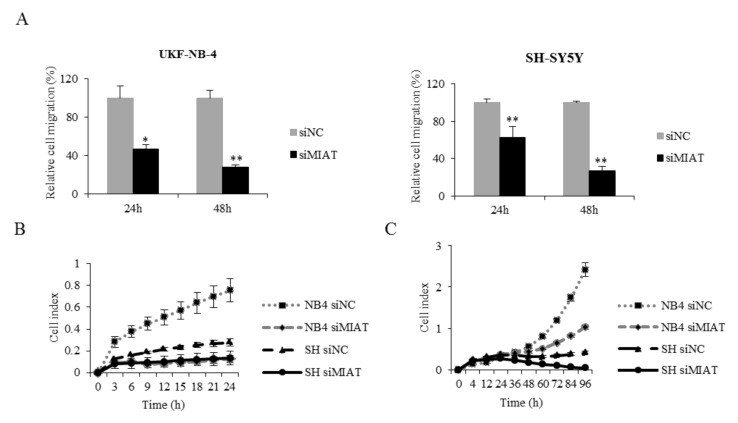
MIAT downregulation decreases UKF-NB-4 and SH-SY5Y cell migration. (**A**) Statistical chart for the scratch-wound gap of UKF-NB-4 and SH-SY5Y cells shows a significant decrease in the migration of MIAT knockdown cells. (**B**) Migration was analyzed using the xCELLigence system after MIAT downregulation. (**C**) Proliferation was analyzed using the xCELLigence system after MIAT downregulation. Data are shown as mean ± SD, * *p* < 0.05, ** *p* < 0.01 (ANOVA with post hoc Tukey HSD Test).

**Figure 6 ijms-22-03393-f006:**
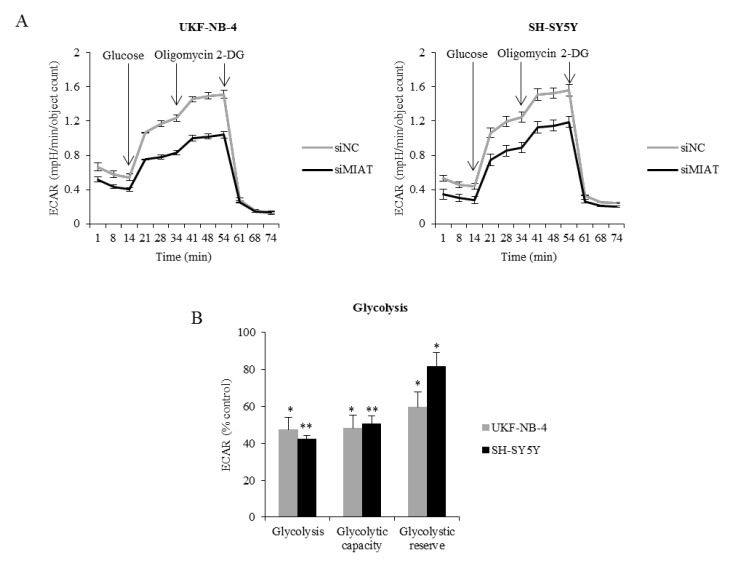
MIAT knockdown decreases glycolytic function. (**A**) The line graph shows changes in ECAR after the addition of glucose, oligomycin, and 2DG. (**B**) The histogram shows glycolysis, glycolytic capacity, and glycolytic reserve in MIAT knockdown cells expressed as % of controls in UKF-NB-4 and SH-SY5Y cells. The graph shows representative data of at least three independent experiments. Values are mean ± SD from independent experiments. ** *p* < 0.01, * *p* < 0.05 as compared to siMIAT/siNC group (ANOVA with post hoc Tukey HSD Test).

**Figure 7 ijms-22-03393-f007:**
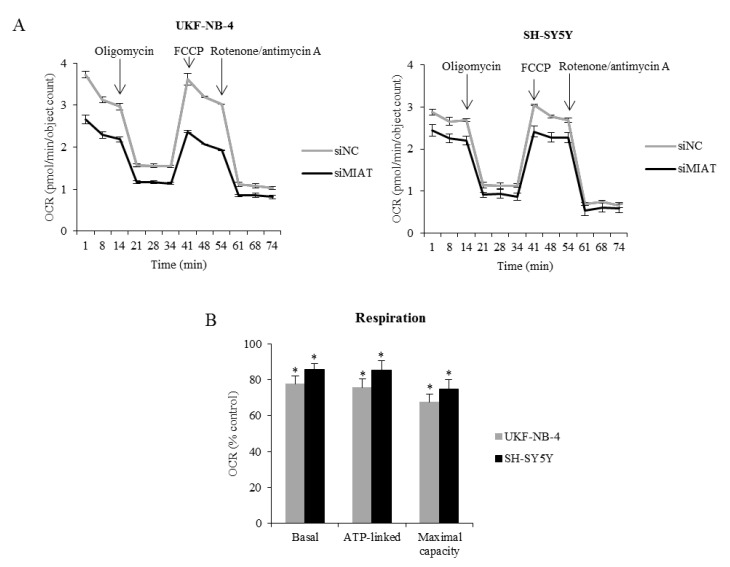
MIAT knockdown decreases respiration. (**A**) The line graph shows changes in OCR after the addition of oligomycin, FCCP, and rotenone/antimycin A. (**B**) Histogram shows basal, maximal, and ATP-associated respiration in MIAT knockdown cells expressed as % of controls in UKF-NB-4 and SH-SY5Y cells. The graph shows representative data of at least three independent experiments. Values are mean ± SD from three independent experiments. * *p* < 0.05 as compared to siMIAT/siNC group (ANOVA with post hoc Tukey HSD Test).

## Data Availability

The datasets generated and/or analyzed during the present study are available from the corresponding author upon reasonable request.

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
