# Peer review of "MIAT Is an Upstream Regulator of NMYC and the Disruption of the MIAT/NMYC Axis Induces Cell Death in NMYC Amplified Neuroblastoma Cell Lines"

_ijms, 2021, doi:10.3390/ijms22073393_

Round 1

Reviewer 1 Report

The authors highlight a putative functional interaction between MIAT an MYC in NB. While some results are potentially interesting, there are some weaknesses in the manuscript in its current form

major points:

1) the authors report that MIAT has not been associated with NB thus far. However, PMID 30836187 is largely focused on the role of MIAT in NB. Furthermore, in PMID 30836187 the authors use siRNAs to KD MIAT in SH-SY5Y cells and look at apoptosis and wound assays. The same experiments are performed in this manuscript. The authors, in my opinion should cite PMID 30836187 and discuss overlap and (most importantly) the different outcomes of their experiments.

2) The authors conclude that MIAT affects MYC expression at transcriptional level. To substantiate this conclusion the authors report that, upon proteasome inhibition, MIAT KD still affects MYC protein and RNA levels.  However, this experiment does not formally prove that MIAT affects the transcription of MYC, as alternative models may explain the same results (e.g. MIAT might promote degradation of MYC mRNA). However, since the underlying mechanism through which MIAT actually affects MYC expression is not investigated, it is questionable whether a conclusive demonstration that MIAT affects MYC at transcriptional level is necessary to substantiate overall conclusions by the authors. I suggest them to either tone down their conclusions on the transcriptional regulation of MYC upon MIAT alteration or, as an alternative, provide more convincing data to support this (i.e. PolII ChIP-qPCR on MYC locus to sustantiate that transcription is affected by MIAT KD).

3) I am not familiar with histochemistry, however the description of the H-score ("Histoscore (H-score) was calculated  by  a  semi-quantitative  assessment  of  the  intensity  of staining  and  the  percentage of MIAT  positive  cells  in  FFPE  NBL  tissues,  staining  by RNA-ISH  method.  The  range of possible scores was from 0 to 300") leaves many open questions: a) how was "semi-quantitative assessment of intensity" performed (equipment, method)?. b) how was this value combined with the % of positive cells? c) how was set the (arbitrary?) range 0-300 of the resulting score?

minor points:

"Western blot analysis was performed using a standard technique. " The authors should provide a reference to a published paper with more details or list the reagents used (methods of protein extraction w/ lysis buffer recipe, protein quantification method, pre-cast gels, membrane, wet/semi dry transfer, ...)

Neuroblastoma is generally referred to as NB, not NBL

"Test was utilized when comparing the situations." I would change the word "situations" with "experimental conditions", "treatments" or other synonym

Author Response

Dear rewiever,

we thank you for your careful review and comments.

major points:

1) the authors report that MIAT has not been associated with NB thus far. However, PMID 30836187 is largely focused on the role of MIAT in NB. Furthermore, in PMID 30836187 the authors use siRNAs to KD MIAT in SH-SY5Y cells and look at apoptosis and wound assays. The same experiments are performed in this manuscript. The authors, in my opinion should cite PMID 30836187 and discuss overlap and (most importantly) the different outcomes of their experiments.

Response: Citation of publication number PMID 30836187 was added and disscused. The main finding of our publication is the importance of MIAT for NBL, especially with NMYC amplification. In particular, the experiments performed on the SH-SY5Y lines should indicate the difference between NMYC amplified and non-amplified cell lines. The experiment's different outcomes in the mentioned publication may be using a different method for detection of apoptosis and different experimental conditions. Moreover, we confirmed low % apoptosis as well on second non-amplified line. In our case, the % of non-apoptotic cells decreased in units of percent, and this change was not significant. They detected still a relatively low rate of apoptosis, approximately 10% compared to the control. We think that differences in apoptosis rate between two laboratories and two different methods about approximately 10% are not so high.

2) The authors conclude that MIAT affects MYC expression at transcriptional level. To substantiate this conclusion the authors report that, upon proteasome inhibition, MIAT KD still affects MYC protein and RNA levels.  However, this experiment does not formally prove that MIAT affects the transcription of MYC, as alternative models may explain the same results (e.g. MIAT might promote degradation of MYC mRNA). However, since the underlying mechanism through which MIAT actually affects MYC expression is not investigated, it is questionable whether a conclusive demonstration that MIAT affects MYC at transcriptional level is necessary to substantiate overall conclusions by the authors. I suggest them to either tone down their conclusions on the transcriptional regulation of MYC upon MIAT alteration or, as an alternative, provide more convincing data to support this (i.e. PolII ChIP-qPCR on MYC locus to sustantiate that transcription is affected by MIAT KD).

Response: It has been fixed in the manuscript.

3) I am not familiar with histochemistry, however the description of the H-score ("Histoscore (H-score) was calculated  by  a  semi-quantitative  assessment  of  the  intensity  of staining  and  the  percentage of MIAT  positive  cells  in  FFPE  NBL  tissues,  staining  by RNA-ISH  method.  The  range of possible scores was from 0 to 300") leaves many open questions: a) how was "semi-quantitative assessment of intensity" performed (equipment, method)?. b) how was this value combined with the % of positive cells? c) how was set the (arbitrary?) range 0-300 of the resulting score?

Response: Firstly, the level of staining intensity (0, 1+, 2+ or 3+) is determined for each cell in the three fields of view. The percentage of cells at each staining intensity level is calculated, and finally, an H-score is assigned using the following formula:

H=[1 × (% cells 1+) + 2 × (% cells 2+) + 3 × (% cells 3+)]

The final score, ranging from 0 to 300, gives more relative weight to higher-intensity of RNA level in a given tumor sample.

Example of color intensity rating for MIAT.

minor points:

"Western blot analysis was performed using a standard technique. " The authors should provide a reference to a published paper with more details or list the reagents used (methods of protein extraction w/ lysis buffer recipe, protein quantification method, pre-cast gels, membrane, wet/semi dry transfer, ...)

Response: It has been fixed in the manuscript.

Neuroblastoma is generally referred to as NB, not NBL

Response: The established abbreviation for neuroblastoma is both NB and NBL. Examples of publications designating neuroblastoma as NBL: PMID: 29445162, PMID: 31612488, PMID: 32886453 etc.

"Test was utilized when comparing the situations." I would change the word "situations" with "experimental conditions", "treatments" or other synonym

Response: It has been fixed in the manuscript.

Reviewer 2 Report

The manuscript entitled ” MIAT Is an Upstream Regulator of NMYC and Disruption of MIAT/NMYC Axis Induces Cell Death in NMYC Amplified Neuroblastoma Cell Lines” by Dr. Hrabeta and colleagues report that an association between long non-coding RNA (lncRNA) myocardial infarction associated transcript (MIAT) and NMYC amplification in NBL cell lines, and MIAT expression in NBL tissue samples as well as suggesting that MIAT might be a potential therapeutic target, especially for those with NMYC amplification.

This manuscript result can support their hypothesis well and well organized. According to author results that MIAT is very efficient therapeutic target for neuroblastoma.

However, in its present form the manuscript raises several questions. The authors should explain and address these questions.

Critical major issues to be addressed:

  1. Can author prove whether this phenomenon is siRNA’s off-target effects? Author needs to do some experiments using siRNA resistance reconstitution.
  2. In Fig. 4, don’t have any G1 cells? G0 population is too high even siNC cells.

Minor corrections:

  1. need to show P-value
  2. need to include 0h samples in Fig. 2B, 5A
  3. need to match viability cells to 100% in Fig. 2C

Author Response

Dear rewiever,

we thank you for your careful review and comments.

Major points:

  1. Can author prove whether this phenomenon is siRNA’s off-target effects? Author needs to do some experiments using siRNA resistance reconstitution.

Response: siRNAs purchased from Dharmacon were of ON-TARGETplus quality and were also SMARTpools of 4 transcriptional variants. ON-TARGETplus siRNA significantly reduces off-target effects while maintaining high levels of target gene knockdown. Pooling of four siRNA duplexes significantly reduces off-target effects of individual siRNAs, while maintaining high levels of target silencing. This should therefore lead to a minimum of off-target effect.

  1. In Fig. 4, don’t have any G1 cells? G0 population is too high even siNC cells.

Response: Thank you for this question. The method does not separate G0 and G1 phases. So it is G0/G1 phase. It has been fixed in the manuscript.

Minor corrections:

  1. need to show P-value

Response: Thank you for this correction. We added the missing P-value in Fig. 2A.

  1. need to include 0h samples in Fig. 2B, 5A

Response: We were not measured non-apoptotic cells 0h after MIAT silencing because there is no effect of silencing. Relative cell migration in figure 5A was calculated as , where A is the area of the initial scratch wound, B is the area of the scratch wound at time 24 or 48 h. So for 0h was relative migration 0%.

  1. need to match viability cells to 100% in Fig. 2C

Response: Viability was matched to untreated cells without silencing, that represented 100%.